# PPARγ-Induced Global H3K27 Acetylation Maintains Osteo/Cementogenic Abilities of Periodontal Ligament Fibroblasts

**DOI:** 10.3390/ijms22168646

**Published:** 2021-08-11

**Authors:** Hang Yuan, Shigeki Suzuki, Shizu Hirata-Tsuchiya, Akiko Sato, Eiji Nemoto, Masahiro Saito, Hideki Shiba, Satoru Yamada

**Affiliations:** 1Department of Periodontology and Endodontology, Tohoku University Graduate School of Dentistry, Sendai 980-8575, Japan; hang.yuan.r2@dc.tohoku.ac.jp (H.Y.); aki-ko.sato.t4@dc.tohoku.ac.jp (A.S.); e-nemoto@umin.ac.jp (E.N.); satoruy@tohoku.ac.jp (S.Y.); 2Department of Biological Endodontics, Graduate School of Biomedical and Health Sciences, Hiroshima University, Hiroshima 734-8553, Japan; shtsuchiya@hiroshima-u.ac.jp (S.H.-T.); bashihi@hiroshima-u.ac.jp (H.S.); 3Department of Restorative Dentistry, Tohoku University Graduate School of Dentistry, Sendai 980-8575, Japan; masahiro.saito.c5@tohoku.ac.jp

**Keywords:** periodontal ligament, PPARγ, histone acetylation, osteogenic differentiation

## Abstract

The periodontal ligament is a soft connective tissue embedded between the alveolar bone and cementum, the surface hard tissue of teeth. Periodontal ligament fibroblasts (PDLF) actively express osteo/cementogenic genes, which contribute to periodontal tissue homeostasis. However, the key factors maintaining the osteo/cementogenic abilities of PDLF remain unclear. We herein demonstrated that PPARγ was expressed by in vivo periodontal ligament tissue and its distribution pattern correlated with alkaline phosphate enzyme activity. The knockdown of PPARγ markedly reduced the osteo/cementogenic abilities of PDLF in vitro, whereas PPARγ agonists exerted the opposite effects. PPARγ was required to maintain the acetylation status of H3K9 and H3K27, active chromatin markers, and the supplementation of acetyl-CoA, a donor of histone acetylation, restored PPARγ knockdown-induced decreases in the osteo/cementogenic abilities of PDLF. An RNA-seq/ChIP-seq combined analysis identified four osteogenic transcripts, RUNX2, SULF2, RCAN2, and RGMA, in the PPARγ-dependent active chromatin region marked by H3K27ac. Furthermore, RUNX2-binding sites were selectively enriched in the PPARγ-dependent active chromatin region. Collectively, these results identified PPARγ as the key transcriptional factor maintaining the osteo/cementogenic abilities of PDLF and revealed that global H3K27ac modifications play a role in the comprehensive osteo/cementogenic transcriptional alterations mediated by PPARγ.

## 1. Introduction

Periodontitis is a lifestyle disease that is characterized by the breakdown of periodontal tissues, which comprise the alveolar bone, periodontal ligament tissue, cementum, and gingiva [1]. More than 10% of the world population have periodontal pockets due to the loss of periodontal tissue [2]. Periodontal tissue homeostasis is maintained by epithelial attachments consisting of the junctional epithelium and, in the root area, by connective attachments comprising collagen fibers [3]. Periodontal ligament tissue is an approximately 0.4-mm-wide fibrous tissue between the cementum and alveolar bone that anchors a tooth in the alveolar socket [4]. Numerous collagen fibers called Sharpey’s fibers, which are the terminal ends of the principal fibers (of the periodontal ligament) that insert into the cementum, are the main component of connective attachments. Since fibroblasts in periodontal ligament tissue (PDLF: periodontal ligament fibroblasts) continuously supply collagen and related molecules to maintain Sharpey’s fibers [4], their activation represents an approach for maintaining periodontal homeostasis and preventing periodontal tissue breakdown [5]. Similar to other organs and tissues, PDLF are the predominant cell type in periodontal ligament tissue [6]. Stem cells surrounding peripheral blood, macrophages, and epithelial cells in the epithelial rests of Malassez are also residential cell types in periodontal tissue [7,8]. As stem cells in periodontal ligament tissue possess higher proliferative and differentiative abilities than PDLF [9], stem cells in periodontal ligament tissue are a useful cell source for periodontal regeneration. However, similar to other tissues and organs, the number of stem cells in periodontal ligament tissue is limited [10]. 

Cell lineage analyses using single cell RNA-seq revealed the diversity of fibroblasts and subgroup-distinct functions [11,12]. Even though the diversity of PDLF currently remains unclear, they have been characterized by two major factors: (1) retained ability to differentiate into osteo/cementogenic cells [13], and (2) the aggressive synthesis and secretion of collagen type 1 and the molecules required for collagen fibrillogenesis in order to compensate for the fast turnover of collagen fibers in periodontal tissue, which is essential for maintaining connective attachments [14]. Despite the lack of evidence to show whether PDLF alone or PDLF and stem cells contribute to periodontal tissue homeostasis, the pivotal roles of PDLF have been validated by the conditional knockout mice of *Wntless* under Osteocalcin-cre recombinase [15]. The secretion of Wnt proteins, such as Wnt3a and Wnt5a, which are inducers of osteo/cementoblastic differentiation [16,17], was eliminated in the *Osteocalcin*-expressing cells of conditional knockout mice of *Wntless* [15]. *Osteocalcin* was shown to be specifically expressed in cells differentiating towards or differentiated osteo/cementoblastic lineage cells, but was negligible in stem cells [18]; however, it was expressed in PDLF [19]. In the periodontal tissue of conditional knockout mice of *Wntless*, the periodontal ligament and cementum were wider, the alveolar bone was narrower, and the expression of osteogenic markers, such as *Alp*, *Runx2*, and *Osterix*, was down-regulated [15]. 

Peroxisome proliferator-activated receptor (PPARγ) is a nuclear receptor that plays a role in energy metabolism, such as glucose and lipid metabolism, by directly regulating numerous metabolic genes [20]. Long-chain fatty acids, 15-deoxy-Δ12,14-PGJ2, oxidized LDL, such as 9-hydroxyoctadecadienoic acid (9-HODE) and 13-HODE, and their metabolites are endogenous agonists of PPARγ [21]. PPARγ exerts anti-inflammatory effects in immunological cells [22] and osteoblastic cells [23] as well as anti-osteoclastogenic effects in periodontal tissue [24]. PPARγ is the master regulator of bone marrow mesenchymal stem cells (BMMSC) for differentiation towards adipogenic cells and inhibits differentiation towards an osteogenic cell lineage [25]. Thiazolidinedione compounds (TZDs) are exogenous agonists of PPARγ and are administered to patients with type II diabetes to improve insulin resistance [26]. TZDs promote global histone acetylation mainly by suppressing the activities of histone deacetylases (HDACs), which remove the acetyl group from the lysine residue [27]. A previous study demonstrated that a HDAC inhibitor suppressed alveolar bone loss in an experimental periodontitis model [28] and induced osteogenic differentiation by expanding the active chromatin region marked by the hyper-acetylation of histone 3 (H3) in vitro [29]. Therefore, PPARγ, a target of TZDs, is presumed to play a pivotal role in maintaining PDL homeostasis.

The present study investigated whether alterations in PPARγ activity by TZDs and the suppression of endogenous PPARγ expression in PDLF modulates osteogenic and ECM-related gene expression. In addition, global epigenetic changes in H3K27ac induced by the suppression of PPARγ were clarified to reveal the underlying mechanisms whereby PPARγ maintains the osteo/cementogenic abilities of PDLF.

## 2. Results

### 2.1. PPARγ Expression in PDL Tissue and PDLF

To establish whether PPARγ is present in periodontal tissue, the demineralized maxilla from three-month-old male mice was stained with the α-PPARγ antibody and positive staining was detected in PDL tissue. These tissues were rich in collagen, similar to the alveolar bone and dentin visualized using Masson’s trichrome stain. Alkaline phosphatase (ALP)-positive cells (brown color) were detected in PDL tissue and dental pulp tissue. PPARγ expression was negligible in epithelial gingival tissue and osteocytes (Figure 1A). Higher magnified images of the root part revealed that PPARγ was widely expressed in PDL tissue (Figure 1B). ALP staining showed high ALP activity was widely detected in PDL especially along with the alveolar bone surface. qPCR analyses revealed that the four PDLF cell types expressed higher levels of *PPARγ*, *PLAP-1/ASPN*, *COL1A1*, *OMD*, and *RUNX2* than BMMSC, with the exception of *OMD* expression in PDLF-2 and *RUNX2* expression in PDLF-3 being as low as that in BMMSC (Figure 1C). 

### 2.2. Ligand-Dependent Modulation of PPARγ Activity Positively Correlates with the Osteo/Cementogenic Potential of PDLF

To clarify whether the exogenous addition of PPARγ agonists modulates the osteo/cementogenic potential of PDLF, PDLF-1 were cultured long-term in induction medium in the presence of several exogenous PPARγ agonists, such as pioglitazone, troglitazone, rosiglitazone, and nTZDpa (5 μM). PDLF-1 stimulated with these PPARγ agonists exhibited higher ALP activity levels than cells treated with DMSO on days 6 and 9 (Figure 2A). The addition of these PPARγ antagonists did not affect cellular proliferative properties examined by MTT (Figure 2B). The addition of pioglitazone, troglitazone, and rosiglitazone resulted in significant increases in *ALPL* expression on day 6, while the addition of pioglitazone and troglitazone also increased *COL1A1* expression (Figure 2C). The addition of PPARγ agonists consistently led to significantly large quantities of calcium deposition, identified by Alizarin red S staining (Figure 2D). These results indicated that PPARγ activity correlated with the osteo/cementoblastic potential of PDLF-1. Furthermore, a 48-h culture of PDLF-1 with these PPARγ agonists in induction medium induced the global acetylation of H3K27ac (Figure 2E).

### 2.3. PDLF Lose Osteo/Cementogenic Gene Expression Due to the Suppressed Expression of PPARγ

The subcellular fractionation of PDLF-1 and PDLF-2 revealed predominant PPARγ expression in the nuclear fraction than in the cytoplasmic fraction (Figure 3A). HSP90α and HDAC1 were used as markers for the cytoplasmic and nuclear fractions, respectively. Then, PDLF-1 and PDLF-2 were transfected with siRNA for *PPARγ* (si-*PPARγ*) or control siRNA (si-control), and qPCR analyses revealed that *PPARγ* expression levels in cells transfected with si-*PPARγ* were 10.6% and 12.1% of expression levels in cells transfected with si-control, respectively (Figure 3B). PPARγ protein expression was also evidently suppressed in PDLF-1 and PDLF-2 following the transfection of si-*PPARγ* (Figure 3C). GAPDH was used as the loading control to validate the amount of protein loaded onto the gel. The suppression of endogenous PPARγ decreased ALP activity in PDLF-1 and PDLF-2 (Figure 3D). The significant inhibition of ALP activity was observed from day 3 and persisted until day 12. The number of cells was not markedly increased or decreased by the transfection of si-*PPARγ* (Figure 3D lower graphs). Changes in the transcription of known osteo/cementogenic differentiation markers, such as *ALPL*, *COL1A1*, *BGLAP*, *PLAP-1/ASPN*, *OMD*, were enumerated. The expression levels of these markers on day 6 were significantly lower in PDLF-1 and PDLF-2 transfected with si-*PPARγ* than in those transfected with si-control (Figure 3E). Among these osteo/cementogenic differentiation markers, *ALPL* and *OMD* that code for the proteins having osteo-inductive functions are significantly increased in BMMSC transfected with si-*PPARγ* on day 12. The suppression of calcium deposition in PDLF-1 and PDLF-2 by the transfection of si-*PPARγ* was consistently visualized by Alizarin red S staining (Figure 3F). 

As shown in Appendix A Figure A1, to validate the effects of the knockdown of PPARγ on the osteo/cementogenic abilities of PDLF, PDLF-1 were transfected with three other independent siRNAs for PPARγ to exclude non-specific targeting effects. PDLF-1 transfected with three independent siRNAs showed lower PPARγ levels than PDLF-1 transfected with si-control (Figure A1A). Cells transfected with si-*PPARγ*-2, si-*PPARγ*-3, and si-*PPARγ*-4 exhibited significantly lower ALP activity levels than those transfected with si-control, even from day 0, which was identical to 24 h after transfection. The number of cells remained unchanged (Figure A1B) and Alizarin red S staining showed lower amounts of calcium deposition in cells transfected with si-*PPARγ*-2, si-*PPARγ*-3, and si-*PPARγ*-4 than in those transfected with si-control (Figure A1C). 

### 2.4. The Knockdown of PPARγ Inhibits the Global Acetylation of H3K9 and H3K27

Since PPARγ is required for PDLF to retain gene expression related to ECM and osteo/cementogenic abilities as well as the agonistic effects of TZDs to induce global H3K27 acetylation (Figure 2), we hypothesized that the suppression of PPARγ may result in epigenetic whole-genomic alterations. Therefore, we investigated whether the knockdown of PPARγ influenced global histone modifications in PDLF-1 (Figure 4). The suppression of PPARγ significantly decreased H3K27ac and H3K9ac, the markers for active chromatin, with a concomitant and significant increase in H3K27me3, a marker for inactive chromatin. 

### 2.5. The Knockdown of PPARγ Comprehensively Alters Gene Expression Profiles

To examine comprehensive gene expression changes induced by the knockdown of PPARγ, PDLF-1 were transfected with si-control or si-*PPARγ* for 24 h, cultured for 6 days in osteo/cementogenic-inducing medium, and RNA was collected before and after osteo-induction (described as “day 0” and “day 6”, respectively, in Figure 5A). RNA-seq analyses of day 0 and 6 samples of control siRNA-treated PDLF-1 showed that 3945 and 5655 transcripts (threshold = 2.0) were up- and down-regulated, respectively, by the 6-day long-term culture (Figure 5A: upper scatterplot). Each transcript spot was colored with a gradient in proportion to its expression ratio (day 6/day 0) with a darker red color for spots having a higher ratio and a darker green color for those with a lower ratio. The middle color was set as white. In this coloring rule, the biological functions of each transcript were not taken into consideration. Fixed colored transcript spots were plotted by their expression levels in PDLF-1 treated with control siRNA (x-axis) and with siRNA for PPARγ (y-axis) on day 6 (Figure 5A: lower scatterplot). On day 6, the expression of green spots (suppressed genes during the long-term culture) was slightly higher in PDLF-1 transfected with si-PPARγ, and that of red spots (inducible genes during the long-term culture) was slightly higher in PDLF-1 cells transfected with si-control. Figure 5B shows the results of gene ontology analyses. The upper rank of the pathway analyses of enhanced biological processes on day 6 relative to that on day 0 (si-control-transfected cells) revealed that genes functionally related to the ECM, such as collagen synthesis and trimerization, proteoglycans, and fibers, were selectively increased by the 6-day culture in induction medium (Figure 5B). The upper rank of pathway analyses of enhanced pathways on day 6 relative to that on day 0 (si-control-transfected cells) showed that “Endochondral Ossification” was ranked third (Figure 5C). This result indicated that osteogenic differentiation activity was enhanced by the 6-day culture in induction medium, as expected. The upper rank of pathway analyses of biological processes suppressed by si-*PPARγ* on day 6 showed that five out of the seven upper ranked biological processes were related to ECM organization and metabolism (Figure 5D). Moreover, the upper rank of suppressed pathways by si-*PPARγ* on day 6 showed that “Endochondral Ossification” was ranked first (Figure 5E). These results suggested that the knockdown of PPARγ was sufficient to eliminate osteo/cementogenic and ECM-organizing potentials by comprehensive transcriptional dysregulation in PDLF.

### 2.6. The Whole-Genomic Identification of H3K27ac Peaks Reveals the Enrichment of RUNX2-Binding Sites in Peaks Retained by PPARγ

PDLF-1 were transfected with si-control or si-*PPARγ* and ChIP-seq was conducted to reveal whole-genomic alterations in H3K27ac modifications by the knockdown of *PPARγ*. The ‘getDifferentiationPeaks.pl’ command in Homer identified more than 60,000 H3K27ac-enriched peaks (si-control-transfected cells: 65,536 peaks and si-*PPARγ*-transfected cells: 61,205 peaks) using input as the background control and, among these, 362 peaks were identified as unique peaks in si-control-transfected cells (si-control unique peaks), 65,173 as common peaks (common peaks), which possessed equivalent read depths in PDLF-1 transfected with si-control and si-*PPARγ*, and 1 as a unique peak in si-*PPARγ* -transfected cells (si-*PPARγ* unique peaks) (Figure 6A). Successful classification was validated by histograms and heatmaps of each duplicate sample (Figure 6A). Among 362 si-control unique peaks, four neighboring gene loci, the transcript functions of which were positively linked with ossification and transcript expression levels was decreased by si-*PPARγ* on day 0 (RNA-seq data set used in Figure 5), were visualized (Figure 6B). qPCR analyses confirmed significantly reduced expression of *SULF2*, *RCAN2*, *RGMA*, and *RUNX2* by si-*PPARγ* at day 0 (Figure 6B). The acetylation of H3K27 in the promoter region (red background) directly influenced the gene expression of sulfatase 2 (*SULF2*) and regulator of calcineurin 2 (*RCAN2*). Furthermore, changes in the acetylation of H3K27 in the distal enhancer of repulsive guidance molecule BMP co-receptor A (*RGMA*) and *RUNX2* were identified. The expression of *SULF2*, *RCAN2*, *RGMA*, and *RUNX2* were down-regulated by si-*PPARγ* in PDLF-2, PDLF-3, and PDLF-4 as similar to PDLF-1 (Figure 6C). Therefore, the expression of these genes were selectively retained by PPARγ-H3K27ac axis in PDLFs. Troglitazone, which exerted the strongest effects on calcium deposition after a long-term culture in induction medium, as shown in Figure 2, significantly induced the gene expression of *SULF2*, *RCAN2*, and *RUNX2* and slightly increased that of *RGMA* on day 6 (Figure 6D). 

The appearance rates of the PPARγ-binding element (PPARE-BE), PPARα-binding element (PPARα-BE), RUNX2-binding element (RUNX2-BE), Smad4-binding element (Smad4-BE), LEF1-binding element (LEF1-BE), and TEAD-binding element (TEAD-BE) in ‘si-control unique peaks’ and ‘common peaks’ were examined (Figure 6E). PPARE-BE (*p* = 0.39), PPARα-BE (*p* = 0.45), Smad4-BE (*p* = 0.83), LEF1-BE (*p* = 0.33), and TEAD-BE (*p* = 0.60) equally appeared in ‘si-control unique peaks’ and ‘common peaks’. However, RUNX2-BE was more frequently detected in ‘si-control unique peaks’ than in ‘common peaks’ (*p* = 0.000098).

### 2.7. Sodium Acetate Supplementation Restores si-PPARγ-Suppressed ALP Activity More Effectively Than the HDAC Inhibitor

To identify the key molecules or intermediate products that organize the si-*PPARγ*-induced suppression of osteo/cementogenic abilities, PDLF-1 transfected with either si-control or si-*PPARγ* were cultured in induction medium for 6 days in the presence of the intermediate products of metabolic pathways. The intermediate products of the TCA cycle, such as isocitric acid, malic acid, and oxaloacetic acid, slightly increased ALP activity in PDLF-1 cells transfected with si-control, whereas none of these restored si-*PPARγ*-inhibited ALP activity (Figure 7A). 

PDLF-1 were then stimulated with sodium acetate, which is rapidly converted to acetyl-CoA and used as the acetyl donor for histone acetylation by histone acetyltransferases, or the HDAC1 inhibitor parthenolide, which inhibited de-acetylation by HDAC1. The supplementation of sodium acetate or parthenolide did not induce ALP activity in PDLF-1 transfected with si-control, whereas that of sodium acetate, but not parthenolide, completely restored decreases in ALP activity in PDLF-1 transfected with si-*PPARγ* (Figure 7B). The ALP activity levels of si-*PPARγ*-treated cells supplemented with 0.5, 1, and 5 mM of sodium acetate on days 6 and 9, were significantly higher than those of si-*PPARγ*-treated cells in the absence of sodium acetate (Figure 7B). Furthermore, calcium deposition of si-*PPARγ*-treated cells was significantly enhanced by acetate (5 mM) supplementation (Figure 7C). 

## 3. Discussion

The present study demonstrated that PPARγ was expressed in PDL tissue in vivo and PDLF in vitro, and that PPARγ maintained an active chromatin status marked with H3K27ac, particularly in the chromatin area in which RUNX2-binding sites were significantly enriched. Furthermore, the exogenous agonistic effects of PPARγ enhanced the osteo/cementogenic abilities of and global acetylation in PDLF.

PDL plays a role in the turnover of the alveolar bone and cementum [6]. The relationship between PPARγ expression and ALP activity implies that PPARγ is indispensable, particularly for alveolar bone turnover (Figure 1B). Furcation-restricted PDL dysregulation phenotypes have been reported in Mohawk homeobox (MKX) knockout mice in which the age-dependent dysregulation of PDL tissue was prominent in the furcation area, but not in the root area [30]. Since the fragment per kb per million (FPKM) of MKX in si-*PPARγ*-transfected cells was 2.72-fold lower than that in si-control-transfected cells on day 0 (RNA-seq data set used in Figure 5A), PPARγ may participate in PDL homeostasis in the furcation area by controlling MKX expression. 

Based on variations in the expression levels of *PLAP-1/ASPN*, a marker of PDL tissue and cells [31], and *RUNX2* (Figure 1), PDLF consist of a heterogenous population [32] and, thus, we herein used PDLF-1 and PDLF-2, which have higher and lower basal ALP activity levels, respectively (Figure 3D). Since the suppression of PPARγ inhibited the osteo/cementogenic abilities of PDLF-1 and PDLF-2, PDLF retain their PPARγ-dependent osteo/cementogenic potential regardless of basal ALP levels. PPARγ has been identified as a key regulator of bone marrow derived mesenchymal stem cells (BMMSC) to induce adipocyte differentiation and inhibit osteogenic differentiation [33]. As expected, the transfection of the same si-*PPARγ* into BMMSC resulted in enhanced osteogenic phenotypes (Figure A2). Therefore, the functions of PPARγ for osteo/cementogenesis in PDLF appear to differ from those in BMMSC. 

Among the 362 peaks identified as ‘si-control unique peaks’, we focused on four transcripts coding known positive regulators of osteogenesis: SULF2, RCAN2, RGMA, and RUNX2 (Figure 6B). A previous study demonstrated that H3K27ac was enriched not only in the promoter region, but also in the distal enhancer/superenhancer region [34]. In the *RUNX2* gene locus, no intergenic H3K27ac peaks were altered by the suppression of PPARγ. In contrast, the distal H3K27ac peaks (Chr 6: 45754312-45759917, 45817138-45818171, and 45899039-45902822) that partially overlapped with the neighboring CLIC5 gene locus were eliminated by the suppression of PPARγ (Figure 6B). *CLIC5* was not expressed in PDLF-1 analyzed using the RNA-seq dataset used in Figure 3. *RUNX2* gene expression is known to be coordinately regulated by proximal and distal enhancers [35]. Therefore, the putative peaks identified may also involve *RUNX2* expression. RUNX2 is one of the key critical transcriptional factors for osteogenesis because global RUNX2 occupancy in osteogenic gene loci has been revealed [36]. SULF2 is a heparan sulfate 6-O-endosulfatase that is highly expressed by hypertrophic chondrocytes, in the endochondral fracture healing spot, and by active osteoblasts [37]. SULF2 has been shown to induce odontogenic differentiation because the desulfation of heparan sulfate proteoglycan (HSPG) by SULF2 liberated Wnt ligands from HSPG and resulted in binding to its receptor [38]. A previous study reported that RCAN2 physiologically inhibited the calcineurin-NFATc1 pathway to enhance osteoblast differentiation and suppressed osteoclast activities; therefore, juvenile RCAN2 null mice had a reduced bone mass [39]. RGMA is a co-receptor and an enhancer of BMP signaling that directly associates with BMP2 and BMP4 and facilitates their binding to activin receptor type IIA [40]. Therefore, the simultaneous inhibition of these four genes may be one of the mechanisms underlying the PPARγ-induced inhibition of the osteo/cementogenic abilities of PDLF. However, the neighboring transcripts of 138 out of 362 si-control unique peaks were categorized as non-coding RNAs (ncRNAs). Several ncRNAs, such as lncRNAs and miRNAs, have been identified as modulators of osteo/cementogenic responses in PDLF [41]. Although these known ncRNAs were not hit as neighboring transcripts of ‘si-control unique peaks’, the exploration of novel functional ncRNAs in our datasets may facilitate a more detailed understating of comprehensive gene expression alterations regulated by the PPARγ-H3K27ac axis.

The osteo/cementogenic abilities of PDLF are regulated by various intracellular signaling pathways, including the RUNX2 activation of osteogenic gene promoters [42], BMPs-Smad signaling [43,44], canonical Wnt-TCF/LEF [45], and the YAP/TAZ-mediated Hippo pathway [46]. As shown in Figure 6E, only RUNX2-BE was enriched in PPARγ-dependent H3K27ac peaks. Therefore, we propose that not only the down-regulated expression of osteogenic transcripts, such as *SULF2*, *RCAN2*, *RGMA*, and *RUNX2*, but also the inaccessibility of the RUNX2 transcriptional complex to its consensus-binding sites are the key molecular mechanisms by which the suppression of PPARγ induces the loss of osteo/cementogenic abilities. Since consensus PPARE- and PPARα-BE were not enriched in ‘si-control unique peaks’ relative to those in ‘common peaks’ (Figure 6E), PPARγ does not appear to directly participate in the local acetylation of histone H3 in 362 ‘si-control unique peaks’. Further analyses, such as Hi-C seq, are required to reveal how peaks with the enrichment of RUNX2-BE are selectively inactivated by the suppression of the PPARγ-H3K27ac axis. 

Intermediate products of the biochemical reactions of metabolic pathways, such as the TCA cycle, fatty acid β-oxidation, and respiratory chain, have been identified as regulators of cellular differentiation through the control of histone modifications [47]. The accumulation of fumarate induced active chromatin markers, such as H3K27 acetylation and H3K4 tri-methylation, at least partially through the inhibition of KDM5 demethylase [48]. Acetyl-CoA and α-ketoglutarate levels have been shown to reflect metabolic activities [49]. Hypermethylation at the CpG island of the promoter region in ECM-related genes was induced by the long-term stimulation of PDLF with LPS [50]. Sodium acetate is immediately converted to acetyl-CoA by ACSS2 and used as an acetyl group for histone acetylation by HATs [51,52,53]. In the present study, the supplementation of acetyl-CoA, but not intermediate products, restored the si-*PPARγ*-dependent inhibition of osteo/cementogenesis (Figure 7). PPARγ is a key modulator of energy metabolism, such as lipid and glucose metabolism [54], while acetyl-CoA is central to lipid and glucose metabolism [55]. However, further studies are warranted to elucidate the molecular mechanisms by which PPARγ controls acetyl group dynamics and H3K27 acetylation in PDLF. 

TZDs are inducers of the differentiation of lineage-specified immature adipocytes into differentiated adipocytes [56]. PDLF have been categorized as a lineage-specified immature cell type [57]. As shown in Figure 5C, ‘Adipogenesis’ was identified in the upper rank of enhanced pathways during osteo/cementogenic differentiation in si-control-transfected cells. Therefore, PDLF and immature adipocytes presumably share differentiation pathways to some extent. TZDs were previously developed as diabetes medicine; however, their current clinical usage is limited. More recent efforts have focused on the drug repurposing of TZDs for cognitive dysfunction [58]. PPARγ cDNA delivery with chitosan gold nanoparticles reduced anti-inflammatory responses on titanium surfaces [59], and these PPARγ-chitosan gold nanoparticles enhanced osteogenic phenotypes by increasing osteogenic gene expression, such as *Runx2* and *Bmp7*, in pre-osteoblastic MC3T3-E1 cells [59,60]. In the present study, nTZDpa, a non-thiazolidine derivative, also exerted positive effects on ALP activity in PDLF (Figure 2). Therefore, the activation of PPARγ may be an ideal strategy for maintaining PDL homeostasis as well as regenerating periodontal tissues.

In conclusion, the present results demonstrated for the first time that PPARγ is a key transcriptional factor in PDLF for the expression of ECM-related and osteo/cementogenic-related genes and also that the agonistic effects of PPARγ enhance osteo/cementogenic abilities. The PPARγ-H3K27ac axis has therapeutic potential for the maintenance of periodontal homeostasis due to preferential epigenetic modifications in the local chromatin region enriched with RUNX2-BE.

## 4. Materials and Methods

### 4.1. Reagents

Rosiglitazone (R2408), pioglitazone (E6910), and nTZDpa (SML0616) were obtained from Sigma-Aldrich (St. Louis, MO, USA). Troglitazone (209-19481) was purchased from FUJFILM Wako Pure Chemical Corporation (Osaka, Japan). Sodium acetate buffer solution (06893-24) was purchased from Nacalai Tesque (Kyoto, Japan).

### 4.2. Histology

Immunostaining, Masson’s Trichrome staining, and alkaline phosphatase (ALP) staining were performed on 5-µm-thick paraffin sections. Three-month-old mice were perfused with phosphate-buffered saline (PBS) to remove circulating blood and then with 4% paraformaldehyde in PBS. The maxilla was removed and fixed in 4% paraformaldehyde in PBS at 4 °C for 24 h. It was then decalcified with 0.134 mol EDTA in PBS at 4 °C for 2 weeks and dehydrated through a graded ethanol series, placed in xylene, and embedded in paraffin. Sections for PPARγ staining were dewaxed and stained with the M.O.M. Immunodetection Kit, Peroxidase, VECTOR < Mouse on Mouse Immunodetection Kit > (PK-2200, Vector Laboratories, Inc., Burlingame, CA, USA) according to the manufacturer’s instructions. Sections were incubated with mouse monoclonal IgG (sc-3877; Santa Cruz Biotechnology Inc., Santa Cruz, CA, USA, diluted in 500 ng/mL) or mouse polyclonal anti-PPARγ (sc-7273; Santa Cruz Biotechnology Inc., diluted in 500 ng/mL) at 4 °C overnight. Immune complexes were visualized using the chromogen substrate 3,3′-diaminobenzidine. Sections were counterstained with hematoxylin and then mounted. Dewaxed sections were also used for Masson’s Trichrome and ALP staining utilizing the TRAP/ALP stain kit (FUJIFILM Wako Chemicals, Osaka, Japan). Histological images were captured using an upright microscope (DM6000 B: Leica, Wetzlar, Germany) with a microscopic digital camera (DP28: Olympus, Tokyo, Japan).

### 4.3. Cell Culture

Human PDLF-1 and human BMMSC were purchased from Lonza Inc. (Walkersville, MD, USA). Three PDLF cell types (PDLF-2, PDLF-3, and PDLF-4) were generated in compliance with the Hiroshima University ethical guidelines for epidemiological research. All experimental procedures were approved by the Committee of Research Ethics of Hiroshima University (Permit Number: E-113). PDLF-2, PDLF-3, and PDLF-4 were isolated from healthy teeth extracted for orthodontic purposes with informed consent. Periodontal tissue was dissected and placed onto a culture dish, and cells that grew from the tissue were used as periodontal ligament cells. PDLF-1 were maintained in SCBM Stromal Cell Growth Basal Medium (CC-3204: Lonza Inc., Basel. Switzerland). PDLF-2, PDLF-3, and PDLF-4 were maintained in low glucose DMEM (Thermo Fisher Scientific, Carlsbad, CA, USA) supplemented with 100 units/mL of penicillin, 100 μg/mL of streptomycin, and 10% FBS. BMMSC were maintained in Mesenchymal Stem Cell Growth Medium (PT-3001: Lonza Inc.). All cells were cultivated at 37 °C under humidified 5% CO_2_ and 95% air atmospheric conditions. Regarding osteogenic induction, PDLF-1, PDLF-2, and BMMSC were cultured in induction medium (low glucose DMEM with ascorbic acid (100 μg/mL) and β-glycerophosphate (10 mM)) with or without various PPARγ agonists, intermediate products of the metabolic pathway, and sodium acetate. Medium was replaced every 3 days. Dexamethasone (10^−8^ M) was also supplemented for a long-term culture of BMMSC.

### 4.4. Quantitative PCR (qPCR) Analysis

RNA was isolated from PDLF-1, PDLF-2, PDLF-3, PDLF-4, and BMMSC with RNAiso plus. Using the ReverTra Ace qPCR RT master mix with a gDNA remover (Toyobo Life Science, Tokyo, Japan), 0.5 μg of total RNA was reverse-transcribed into cDNA using mixed primers. qPCR reactions were prepared with the KAPA SYBR Fast qPCR kit (KAPA BIOSYSTEMS, Woburn, MA, USA). *HPRT* was used as an internal reference control. PCR primer sequences for target genes are shown in Table A1. 

### 4.5. ALP Activity

ALP activity was measured as previously described [61]. It was normalized to cell numbers from a parallel cell culture quantified by the MTT Cell Count Kit (23506-80: Nacalai Tesque). 

### 4.6. Alizarin Red S Staining

PDLF and BMMSC were washed twice with DPBS and fixed with 70% ethanol for 10 min. Fixed cells were washed with dH_2_O and stained with 1% Alizarin red S (pH 4.2) solution. Images of each stained well were converted to a black/white scale for quantification. Regions where the cells were detached from the well were excluded from the analysis. The intensity average of the area containing cells was measured using ImageJ. Since black and white were set as low and high, respectively, in the black/white scale, the value measured was subtracted from the average intensity of an empty well. Normalized Alizarin red staining scores were obtained by multiplying the subtracted average intensity by the area containing cells. 

### 4.7. Transient Transfection of siRNA

RNA sequences for targeting *PPARγ* by siRNA were selected using Enhanced siDirect, a web-based target-specific siRNA design software. Control siRNA was previously described [62]. siRNAs were generated by Sigma-Aldrich. The siRNA sequences of 4 types of siRNAs for *PPARγ* (si-*PPARγ*) and that of control siRNA (si-control) are shown in Table A2. siRNAs were forward-transfected into PDLF-1, PDLF-2, PDLF-3, PDLF-4, and BMMSC at a final concentration of 10 nM using Lipofectamine RNAiMAX reagent and then incubated for 24 h.

### 4.8. Histone Purification

PDLF-1 and PDLF-2 were transfected with si-*PPARγ* or si-control for 24 h. These cells were then cultured in normal growth medium for 24 h and histone proteins were obtained using the EpiQuik Total Histone Extraction Kit (Epigentek, Farmingdale, NY, USA).

### 4.9. Immunoblotting

Reduced samples were loaded onto NuPAGE Bis-Tris (Thermo Fisher Scientific) gels in MOPS buffer and separated proteins were transferred onto a PVDF membrane for immunodetection with anti-PPARγ (C26H12, 1:2000: Cell Signaling Technologies, Danvers, MA, USA), HSP90α (GTX109753: Gene Tex, Irvine, CA, USA, 1:2000), HDAC1 (GTX100513: Gene Tex, 1:2000), GAPDH (GTX100118: Gene Tex, 1:1000), H3K27ac (39134: Active Motif, Carlsbad, CA, USA, 1:2000), H3K9ac (39918: Active Motif, 1:2000), H3k27me3 (C36B11: Cell Signaling Technologies, 1:2000), and Histone H3 (D2B12: Cell Signaling Technologies, 1:2000) antibodies as primary antibodies, and the HRP-conjugated goat anti-rabbit antibody (#7074: Cell Signaling Technologies, 1:2000) as the secondary antibody. 

### 4.10. Subcellular Fractionation

PDLF-1 and PDLF-2 were transfected with si-*PPARγ* or si-control for 24 h. These cells were then cultured in normal growth medium for 24 h and cytoplasmic and nuclear fractions were obtained using NE-PER™ Nuclear and Cytoplasmic Extraction Reagents (Thermo Fisher Scientific). 

### 4.11. RNA-seq

PDLF-1 were transfected with si-*PPARγ* or si-control for 24 h. These cells were then cultured in induction medium for 6 days and total RNA was purified before and after the culture. Purified RNA was DNase-treated and used in RT-qPCR and RNA-seq analyses. Input RNA was converted to cDNA libraries using the NEBNExt poly (A) mRNA magnetic isolation module. Libraries were sequenced using 150-bp paired-end reads on Illumina Novaseq instruments. Library preparation and RNA-seq were performed by Novogene (Beijing, China).

Adapter trimming was conducted with Trim galore version 0.6.6 (http://www.bioinformatics.babraham.ac.uk/projects/trim_galore/ (accessed on 3 May 2021)) with default settings and then aligned to a reference genome (hg38) using HISAT2 version 2.2.1 [63]. Tag directories were generated using ‘makeTagDirectory’ in HOMER [64] and gene expression at exons was quantified with the ‘analyzeRepeast.pl’ command in HOMER using ‘-strand both’ and ‘-count exons’ to identify expressed genes.

### 4.12. Chromatin Immunoprecipitation (ChIP)-seq

PDLF-1 were transfected with si-*PPARγ* or si-control for 24 h and then cultured in normal growth medium for 24 h. Cells used in the ChIP-seq analysis were fixed with formaldehyde and ChIP was performed using the SimpleChIP plus enzymatic chromatin IP kit and magnetic beads (Cell Signaling Technologies) following the manufacturer’s protocol. Equal amounts of chromatin (estimated as chromatin obtained from 4 × 10^6^ cells) were used for the precipitation of protein-DNA complexes with a ChIP-validated anti-H3K27ac antibody (ab4729: Abcam, Cambridge, MA, USA). After reverse-crosslinking with a proteinase K treatment, chromatin was eluted using the ChIP DNA clean and concentrator (Zymo Research, Irvine, CA, USA). Libraries were sequenced using 150-bp paired-end reads on an NovaSeq instrument to a depth of >10 million mapped reads. Sequence library preparation using NEBNExt ChIP-seq for Illumina and library sequences were performed by Novogene. 

Adapter trimming and quality checks were conducted with Trim galore version 0.6.6 (http://www.bioinformatics.babraham.ac.uk/projects/trim_galore/ (accessed on 31 March 2021)) and then aligned to a reference genome (hg38) using Bowtie2 version 2.4.1 [65] with ‘-seed’ parameters to efficiently use multimap reads with assuring reproducibility. Tag directories were generated using the ‘makeTagDirectory’ command in HOMER and ChIP-seq peaks were identified using the ‘getDiffrentialPeaksReplicates.pl’ command in HOMER, specifying ‘-style histone’ to generate a high confidence set of peaks across duplicates. This HOMER command internally calls DESeq2 to calculate the enrichment value for each peak using the individual raw counts and only returns peaks that pass 2-fold enrichment and a FDR-adjusted *p*-value of <0.05. Equal amounts of chromatin DNA from duplicates of si-control-transfected PDLF-1 were mixed to prepare input DNA-1. Similarly, those of si-*PPARγ*-transfected cells were mixed to prepare input DNA-2. The input DNA-1 and input DNA-2 were used as input DNA duplicates.

The ‘si-control unique peaks’, ‘si-PPARγ unique peaks’, and ‘common peaks’ were identified by the ‘getDiffrentialPeaksReplicates.pl’ command in HOMER with the default setting using biological duplicates of si-control tag directories as ‘sample’, that of si-*PPARγ* tag directories as ‘background’, and that of input as ‘input’. Peaks with significantly higher H3K27ac tag densities in si-control-transfected cells than in si-*PPARγ*-transfected cells were defined ‘si-control unique peaks’, while those having significantly higher H3K27ac tag densities in si-PPARγ-transfected cells than in si-control-transfected cells were defined as ‘si-*PPARγ* peaks’. H3K27ac peaks with equivalent tag densities in cells treated with si-control and si-*PPARγ* were defined as common peaks. The threshold for the number of tags needed for a valid peak was selected for a FDR-adjusted *p*-value of <0.05 with more than a 2-fold local enrichment in ‘sample’ than in ‘background’. 

To generate histograms, tags were quantified using the ‘annotatePeaks.pl’ command in HOMER, specifying ‘-size 20000 -hist 50‘. To generate a heat map, tags were calculated with computerMatrix in deeptools [66] and heatmaps were drawn by plotHeatmap in deeptools [66]. To quantify the number of binding elements of various transcriptional factors, such as PPARE, PPARα, RUNX2, Smad4, LEF1, and TEAD, the bind elements in ‘si-control unique peaks’ and ‘common peaks’ were searched using the ‘annotatePeaks.pl’ command in HOMER, specifying ‘-size given’ and ‘-nmotifs’. The number of binding elements in each peak was normalized by the peak size and visualized as the number of peaks per 1 kbp. Regarding visualization in the UCSC Genome browser, BAM files were converted to bigwig files using ‘bamCoverage’ in deeptools with -binsize = 10, minMappingQuality 10 [66].

### 4.13. Accession Numbers

RNA-seq and ChIP-seq data sets have been deposited in NCBI’s GEO with accession number GSE178607.

### 4.14. Statistical Analysis

Statistical analyses were performed by two-tailed unpaired Student’s *t*-tests (Figure 2A,C,D, Figure 3B,D,E, Figure 4, Figure 6D, Figure 7C, Figure A1, Figure A2) and a one-way analysis of variance (ANOVA), followed by Dunnett’s test (Figure 7B).

## Figures and Tables

**Figure 1 ijms-22-08646-f001:**
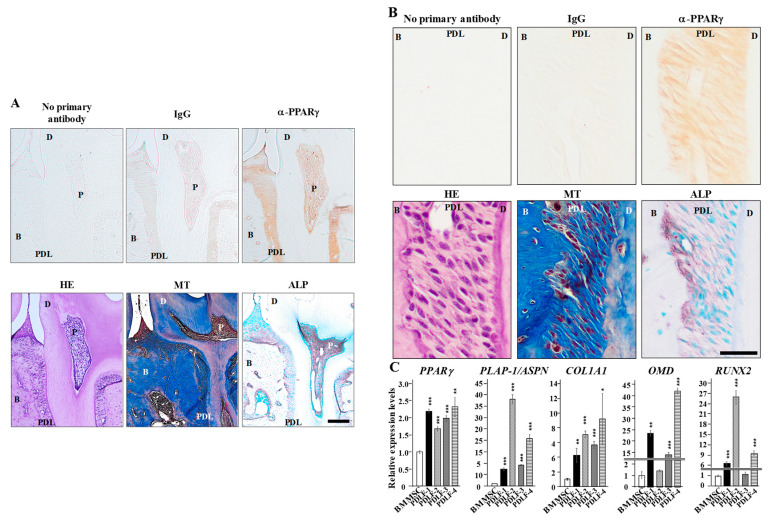
**PPARγ expression in PDL tissue and cells**. (**A**,**B**) Demineralized 3-month-old maxilla sections were stained without the primary antibody or with mouse IgG or α-PPARγ and were also stained with Masson’s Trichrome stain and ALP. (**C**) Total RNAs of PDLF-1, PDLF-2, PDLF-3, PDLF-4, and BMMSC were collected to quantify the expression of *PPARγ*, *PLAP-1/ASPN*, *COL1A1*, *OMD*, and *RUNX2*. *HPRT* was used for normalization. * *p* < 0.05; ** *p* < 0.01; *** *p* < 0.001 significantly higher than BMMSC cells. Scale bars correspond to 200 μm (**A**) and 50 μm (**B**). MT = Masson’s Trichrome staining.

**Figure 2 ijms-22-08646-f002:**
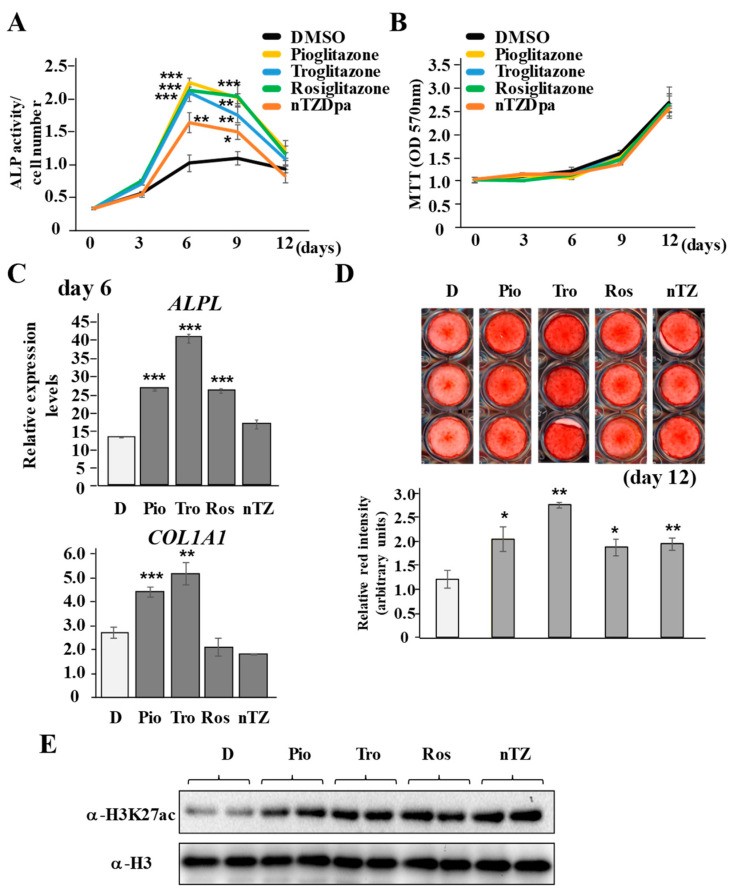
**PPARγ agonists enhance osteo/cementogenic abilities of PDLF-1**. (**A**–**D**) PDLF-1 were cultured in induction medium for a maximum of 15 days. ALP activities were normalized by cell numbers (**A**), cell numbers were enumerated by MTT (**B**), gene expression changes in *ALPL* and *COL1A1* by qPCR (**C**), and calcium deposition were visualized and quantified by Alizarin red S staining (**D**). (**E**) PDLF-1 were cultured in the presence of PPARγ agonists for 48 h and the global acetylation of H3K27 was analyzed by SDS-PAGE using whole histone H3 as the loading control. H3 = histone 3. * *p* < 0.05; ** *p* < 0.01; *** *p* < 0.001 significantly higher than in cells treated with DMSO on the same day. D = DMSO, Pio = Pioglitazone, Tro = Troglitazone, Ros = Rosiglitazone, nTZ = nTZDpa.

**Figure 3 ijms-22-08646-f003:**
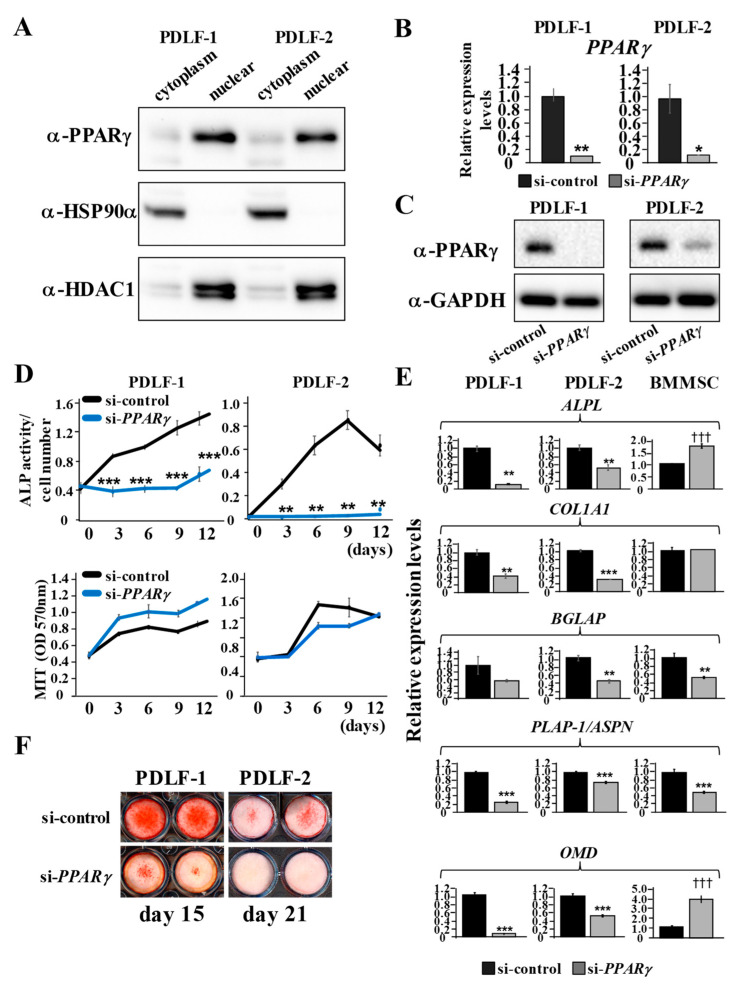
**Suppression of *PPARγ* expression eliminates osteo/cementogenic abilities of PDLF**. (**A**) Cytoplasmic and nuclear fractions were isolated separately from PDLF-1 and PDLF-2 and loaded onto SDS-PAGE gels. (**B**–**F**) PDLF-1 and PDLF-2 were transfected with si-control or si-*PPARγ*. (**B**,**C**) *PPARγ* mRNA expression (**B**) and protein expression (**C**) 24 h after transfection was analyzed using *HPRT* for normalization (**B**) and GAPDH as the loading control (**C**). (**D**) ALP activities were measured and cell numbers were enumerated by MTT every 3 days. ALP activities were normalized by MTT values. (**E**) The gene expression levels of *ALPL*, *COL1A1*, *BGLAP*, *PLAP-1/ASPN*, and *OMD* were examined on day 6 for PDLFs and on day 12 for BMMSC. (**F**) Alizarin red S staining was performed on day 15 for PDLF-1 and on day 21 for PDLF-2. Data represent the mean ± SD of three independent experiments. * *p* < 0.05; ** *p* < 0.01; *** *p* < 0.001 and ††† *p* < 0.001 significantly lower and higher in cells transfected in si-*PPARγ*, respectively.

**Figure 4 ijms-22-08646-f004:**
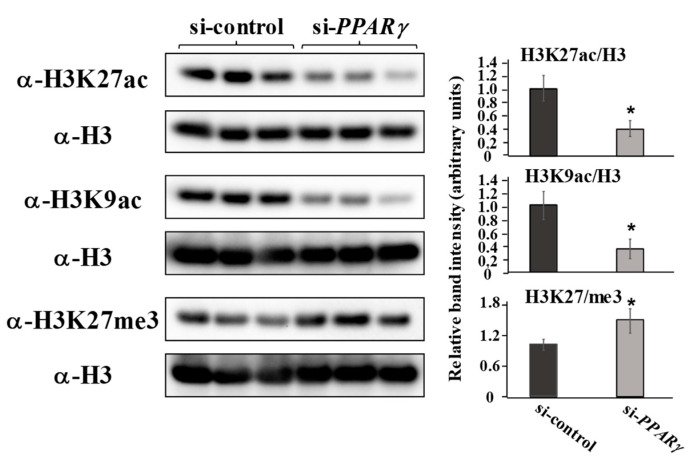
**PPARγ suppression inhibits the global acetylation of H3K9 and H2K27**. PDLF-1 were transfected with si-control or si-*PPARγ* for 24 h and then recovered in normal growth medium for another 24 h. Total histones were collected and histone modifications, such as H3K27ac, H3K9ac, and H3K27me3, were examined using specific antibodies. Band intensities are normalized against those obtained using the antibody for whole histone 3. * *p* < 0.05 significantly lower than in cells transfected with si-control.

**Figure 5 ijms-22-08646-f005:**
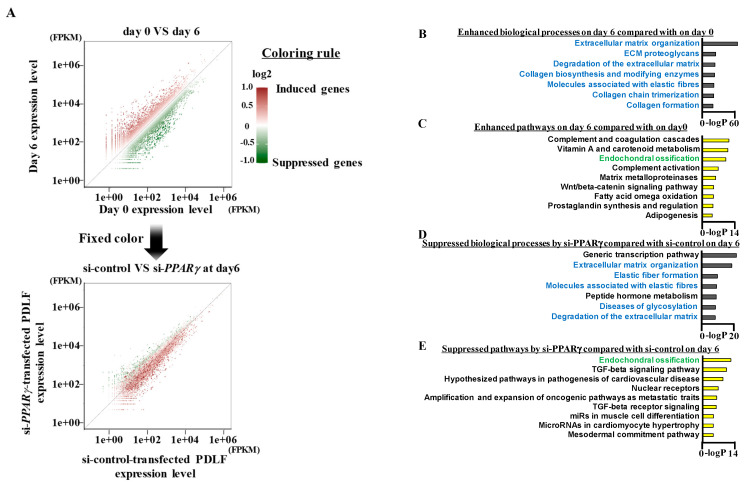
**PPARγ suppression comprehensively reduces the expression of ECM- and ossification-related genes in PDLF**. (**A**) PDLF-1 were transfected with si-control or si-*PPARγ* and then cultured for 6 days in induction medium. Total RNAs were collected before and after the culture and whole genomic transcriptional changes were assessed by RNA-seq. The upper scatterplot shows a comparison before (day 0: x-axis) and after (day 6: y-axis) the long-term culture of si-control-transfected PDLF. The color of each gene spot is linked with a gradient in proportion to its expression ratio (day 6/day 0), with a darker red color for spots having a higher ratio and a darker green color for those with a lower ratio. The middle color is set as white. The lower scatterplot shows a comparison between si-control-transfected PDLF (x-axis) and si-*PPARγ*-transfected PDLF (y-axis). The color of each gene spot was taken over from the upper scatterplot. (**B**–**E**) Pathway analyses of biological processes (**B**,**D**) and wikipathways (**C**,**E**). Enhanced terms on day 6 from those on day 0 (si-control-transfected cells) (**B**,**C**) and suppressed terms by si-*PPARγ* from si-control on day 6 (**D**,**E**).

**Figure 6 ijms-22-08646-f006:**
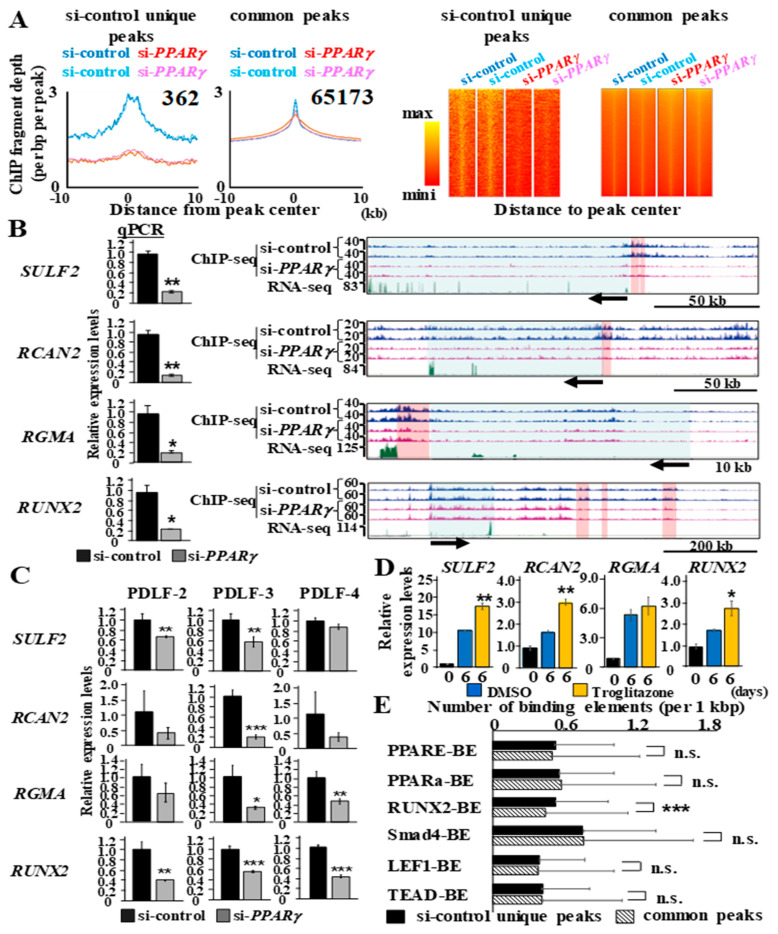
**Local genomic peaks retained by PPARγ include significantly higher numbers of RUNX2-binding sites**. (**A**) Histograms represent the occurrence of H3K27ac peaks within 10 kb of the ‘si-control unique peak’ and ‘common peak’ centers. A heat map comparing the binding of technical duplicate tags from PDLF-1 transfected with either si-control or si-*PPARγ* within 10-kb windows surrounding the ‘si-control unique peak’ and ‘common peak’ centers. (**B**) qPCR analysis is shown as a bar graph. UCSC genomic browser tracks of each gene locus showed a gene locus (blue background) and H3K27ac peaks identified as ‘si-control unique peaks’ (red background). Arrows indicate transcriptional direction. (**C**) PDLF-2, PDLF-3, and PDLF-4 were transfected with si-control or si-*PPARγ* and *SULF2*, *RCAN2*, *RGMA*, and *RUNX2* mRNA expression at 24 h after transfection was analyzed using *HPRT* for normalization. (**D**) PDLF-1 were cultured in induction medium for 6 days in the presence or absence of troglitazone (5 μM) to quantify the expression of *SULF2*, *RCAN2*, *RGMA*, and *RUNX2*. *HPRT* was used for normalization. (**E**) The number of PPARγ-binding elements (PPARE-BE), PPARα-binding elements (PPARα-BE), RUNX2-binding elements (RUNX2-BE), Smad4-binding elements (Smad4-BE), LEF1-binding elements (LEF1-BE), and TEAD-binding elements (TEAD-BE) in ‘si-control unique peak’ and ‘common peak’ were normalized as peaks/1 kbp. The average peak/1 kbp of each BE in ‘si-control unique peaks’ and ‘common peaks’ is shown. * *p* < 0.05; ** *p* < 0.01; *** *p* < 0.001 significantly higher in PDLF-1 treated with troglitazone than in PDLF-1 treated with DMSO on day 6 (**D**) and in ‘si-control unique peaks’ than in ‘common peaks’ (**E**). n.s. = no significant difference.

**Figure 7 ijms-22-08646-f007:**
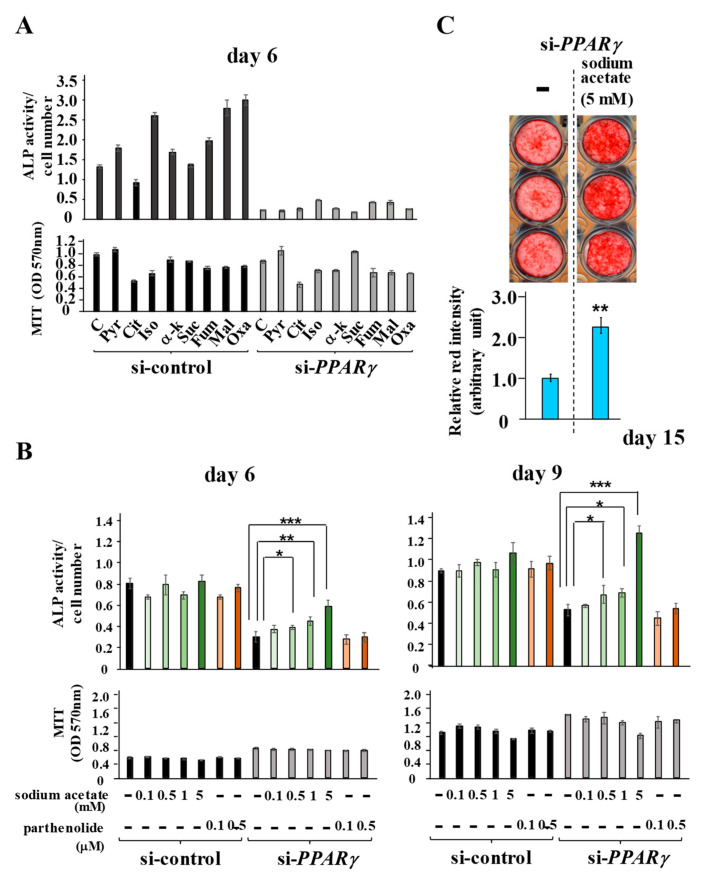
**Sodium acetate supplementation restores si-*PPARγ* -suppressed ALP activity**. (**A**–**C**) PDLF-1 were transfected with si-control or si-*PPARγ* for 24 h and then cultured in induction medium for the indicated periods in the presence or absence of metabolic pathway intermediate products (**A**) and sodium acetate or parthenolide, a HDAC1 inhibitor (**B**). Calcium deposition was visualized and normalized by Alizarin red staining (**C**). H3 = histone 3, Pyr = pyruvic acid, Cit = citric acid, Iso = isocitric acid, α-k = α-ketoglutaric acid, Suc = succinic acid, Fum = fumaric acid, Mal = malic acid, Oxa = oxaloacetic acid. Data represent the mean ± SD of three independent experiments. * *p* < 0.05; ** *p* < 0.01; *** *p* < 0.001 significantly higher than in cells treated with neither sodium acetate nor parthenolide in the same siRNA group on the same day (**B**) and in si-*PPARγ*-transfected cells without sodium acetate supplementation (**C**). n.s. = no significant difference.

## Data Availability

RNA-seq and ChIP-seq data sets have been deposited in NCBI’s GEO with accession number GSE178607.

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
