# Peer review of "PPARγ-Induced Global H3K27 Acetylation Maintains Osteo/Cementogenic Abilities of Periodontal Ligament Fibroblasts"

_ijms, 2021, doi:10.3390/ijms22168646_

Round 1

Reviewer 1 Report

Manuscript ID: ijms-1324489

Title: PPARg-induced global H3K27 acetylation maintains osteo/cementogenic abilities of periodontal ligament fibroblasts

Authors: Hang Yuan, Shigeki Suzuki *, Shizu Hirata-Tsuchiya, Akiko Sato, Eiji Nemoto, Masahiro Saito, Hideki Shiba, and Satoru Yamada

Submitted to section: Molecular Pathology, Diagnostics, and Therapeutics,

General comments:

In the present study, PPARg was detected in mouse PDL fibroblasts (PDLF) by the peroxidase-IHC method. PPARg agonists were shown to increase the osteogenic potential of PDLF, whereas PDLF cells lost their osteogenic mRNA expression after PPARg knockdown using siRNA gene silencing method. PPARg maintains the acetylation status of H3K9 and H3K27, and a donor of histone acetylation improved the PPARg knockdown-induced decrease in the osteogenic potential of PDLF. The osteogenic genes RUNX2, SULF2, RCAN2, and RGMA were enriched in the PPAR-dependent active chromatin region marked by H3K27ac. From these results, the authors concluded that PPARg, as a major transcription factor in PDLF, may regulate the expression of osteogenic genes through epigenetic modifications of H3K27.

Potentially, this an important study, that may identify a novel regulatory function of the PPARg in the expressions of osteogenic genes including RUNX2 in PDLF by acetylation of H3K27. With the exception of the immunohistochemical results, the results obtained by histochemistry, immunoblot and the molecular biology methods are very well presented for a reader.

Specific comments to the authors:

There is no difference between stainings in the nuclei of PDLFs performed with secondary antibody (control) or with PPAR antibody, although the authors have shown strong localization of PPAR in the nucleus of PDLF by immunoblot (Fig. 3A) in cell lines (PDLF). A faint orange staining with the PPAR antibody is seen only in the extracellular region of the PDL, which is different from control staining without primary antibody or with secondary antibody only. However, PPAR should not be present in the extracellular area. In addition, this extracellular orange staining is not a background staining because it is present only in the PDL (in the alveolar bone and dentin, this orange staining is also not present).

Compared with cell lines, cellular protein localizations in tissue sections have very strong significance. Therefore, the specific localization of PPAR in tissue PDLF is of great importance here. In Masson trichrome staining, extracellular blue-stained collagen is seen in PDL, bone, and also in dentin. In Masson trichrome staining, pink stained cytoplasm and brown stained nuclei of osteoclasts are seen in the transition area from PDL to alveolar bone and also in some PDLFs. This shows that the fibroblasts in the PDL with cytoplasm and with nucleus can be seen in this magnification. PPAR should also be clearly visualized in the cytoplasm and especially in the nucleus of PDLF by immunohistochemical methods. Because the antibody is very specific according to the data sheet and these IHC data significantly weaken the importance of this good manuscript, Figure 1A and 1B should be changed with "no primary antibody," "IgG," and "αPPARγ." Therefore, the method of IHC should be modified. Methodically, I can recommend the following:

1) Perform antigen retrieval on paraffin sections.

2) Test antibody dilutions at 1:400, 1:800 and 1:1200.

3) DAB substrate solution can be prepared according to the sensitive concentration (Korkmaz et al., Int. J. Mol. Sci. 2021, 22(2), 539).

4) Counterstaining is not necessary. Therefore, do not perform HE counterstaining.

5) Or the authors can (if they have the appropriate antibodies) perform double immunofluorescence staining with histones H3 and PPARg in mouse tissue sections.

Author Response

We wish to express our appreciation to the first reviewer for their insightful comments on our paper. The comments have helped us significantly improve the paper.

Reviewer #1: Potentially, this an important study, that may identify a novel regulatory function of the PPARg in the expressions of osteogenic genes including RUNX2 in PDLF by acetylation of H3K27. With the exception of the immunohistochemical results, the results obtained by histochemistry, immunoblot and the molecular biology methods are very well presented for a reader.

We strongly appreciate the first reviewer's comment on this point. As the reviewer suggested, we performed immunohistochemical analyses as following the first reviewer`s suggested protocol and revised Fig. 1A and B. In higher magnification of PDL tissue, we found that entire cell bodies including nuclear of PDLF were PPARϒ positive but surrounding ECM area was PPARϒ negative.

We found the error in antibody information. The anti-PPARϒ (sc-7273; Santa Cruz Biotechnology Inc.,) is mouse monoclonal but not rabbit polyclonal. We corrected it in the revised manuscript.  

We hope that the first reviewer approves of this revised version.

Reviewer 2 Report

Line 123: I assime 'antagonists' is a typo.

The three PDLF (-2, -3, -4) cell lines are nor characterized or validated and there is not clear demonstration on morfphological, genetic or expression level of their chracteristics /identity.

A lot of the transfection experiment could be done on any cell line, even HeLa, and could show similar results of PPARg manipulation.

RNA was extracted from the cell lines but only expression data of a few genes are shown. Large profiles would show that they truly cluster together and separately from BMMSC.

The results of the RNA-seq experiment are used for further analyses but only one of the cell lines was used (and even without technical replicates). There is no proof the differentially regulated genes are not random. This should be done on a few PDLFs and overlapping transcripts should be prioritized.

Is there a therapeutic potential for TZds? 

Author Response

We wish to express our appreciation to the second reviewer for their insightful comments on our paper. The comments have helped us significantly improve the paper. In the revised manuscript, we addressed all of your helpful comments as described above. We hope that second reviewer approves of this revised version.

  1. Line 123: I assime 'antagonists' is a typo.

Thank you for your suggestion. However, TZDs are the agonists of PPARϒ and therefore global acetylation of H3K27 was increased by the treatment with TZDs as shown in Fig. 2E. We therefore wish to retain the original text on this point.

  1. The three PDLF (-2, -3, -4) cell lines are nor characterized or validated and there is not clear demonstration on morfphological, genetic or expression level of their chracteristics /identity.

Thank you for your suggestion. We believe that the best PDLF marker is PLAP1/ASPN. In order to emphasize the significantly higher expression of PLAP1/ASPN in PDLFs than that in BMMSC, we revised the bar graphs and the relative expression level of PLAP1/ASPN in BMMSC set as 1. We also conducted statistical analyses, which revealed the significant higher expression of PLAP1/ASPN in PDLFs. Additionally, we also analyzed the expression of OMD (osteomodulin), which is also the maker of PDLF (doi.org/10.1007/s00418-012-0923-6) and found that OMD expression was significantly high in PDLF-1, PDLF-3, and PDLF-4 than in BMMSC. Also it was slightly high in PDLF-2 than in BMMSC. We hope that revised Fig. 1C satisfies the reviewer`s suggestion.

Accordingly, we revised the explanatory sentence (p. 2, lines 81-83) as follow.

" qPCR analyses revealed that the 4 PDLF cell types expressed higher levels of PPARϒ, PLAP-1/ASPN, COL1A1, OMD, and RUNX2 than BMMSC, with the exception of OMD expression in PDLF-2 and RUNX2 expression in PDLF-3 being as low as that in BMMSC (Fig. 1C). "

  1. A lot of the transfection experiment could be done on any cell line, even HeLa, and could show similar results of PPARg manipulation.

We appreciate the reviewer's interest in the effects of PPARϒ overexpression in PDLF. However, the transfection efficiency of plasmid vector into PDLF was apparently low and also the antibiotic selection may lead biased PDLF survival due to the heterogeneity of PDLF even if retro/lenti viral infection system are used for the transfection. In this study, we used various kinds of exogenous agonists of PPARg and therefore we consider that overexpression strategy may not be necessary. Further, in order to demonstrate that TZDs and nTZDpa, a non-thiazolidine derivative, induced osteo/cementogenic abilities of PDLF dependently on PPARϒ, we stimulated PDLF-2 transfected with si-PPARϒ with Pioglitazone and nTZDpa. We found that Pioglitazone and nTZDpa were unable to increase alkaline phosphate (ALP) activity in the cells transfected with si-PPARϒ unlikely to the cells transfected with control siRNA, which showed increased ALP activity at both day 9 and day 12 (Please see the attached word file). We therefore wish to retain the original text on this point.

  1. RNA was extracted from the cell lines but only expression data of a few genes are shown. Large profiles would show that they truly cluster together and separately from BMMSC.

Thank you for your suggestion. The results of the additional work suggested by the reviewer showed revised Fig. 3E. In addition to ALPL, COL1A1, and BGLAP, which were already shown in the original manuscript, PLAP-1/ASPN and OMD expression changes by si-PPARϒ were also analyzed. PLAP-1/ASPN and OMD known to be expressed in PDLF and their expression levels are increased during osteogenic differentiation of PDLF. Since the suppression of PPARϒ lead the decreased expression of ALPL, COL1A1, BGLAP, PLAP1/ASPN, and OMD, we think PPARϒ altered comprehensive gene expression in PDLF-2 as similar to PDLF-1, which were shown in revised Fig. 5A and B. BGLAP and PLAP1/ASPN expression was decreased and OMD was increased in BMMSC. OMD is known to be the accelerator of osteogenesis. Thus, the upregulation of OMD in BMMSC by si-PPARϒ may be one of the key factors of osteo-induced phenotypes in BMMSC, which were shown in revised Appendix Figure 2. 

Accordingly, we revised the explanatory sentence (p. 4, lines 110-114) as follow.

" Changes in the transcription of known osteo/cementogenic differentiation markers, such as ALPL, COL1A1, BGLAP, PLAP-1/ASPN, OMD, were enumerated. The expression levels of these markers on day 6 were significantly lower in PDLF-1 and PDLF-2 transfected with si-PPARg than in those transfected with si-control (Fig. 3E). Among these osteo/cementogenic differentiation markers, ALPL and OMD that code for the proteins having osteo-inductive functions are significantly increased in BMMSC transfected with si-PPARϒ. "

  1. The results of the RNA-seq experiment are used for further analyses but only one of the cell lines was used (and even without technical replicates). There is no proof the differentially regulated genes are not random. This should be done on a few PDLFs and overlapping transcripts should be prioritized.

Thank you for your suggestion. The results of the additional work suggested by the reviewer showed revised Fig. 6C. The identified osteogenic genes (SULF2, RCAN2, RGMA, and RUNX2) in PDLF-1 were also down-regulated by si-PPARϒ in PDLF-2, PDLF-3, and PDLF-4. Therefore, we think the expression of these genes are selectively retained by PPARϒ-H3K27ac axis in PDLFs.

Accordingly, we revised the explanatory sentence (p. 8, lines 164-165) as follow.

"The expression of SULF2, RCAN2, RGMA, and RUNX2 were down-regulated by si-PPARϒ in PDLF-2, PDLF-3, and PDLF-4 as similar to PDLF-1. Therefore, the expression of these genes were selectively retained by PPARϒ-H3K27ac axis in PDLFs. "

  1. Is there a therapeutic potential for TZds?

Thank you for your suggestion. We agree that additional information on in vivo effects of TZDs as the reviewer suggested would be valuable. We are now investigating this point and intend to report it in a later paper.

Round 2

Reviewer 2 Report

The authors took good effort to enhance the manuscript and the work, following the revision, is currently suitable for publication.